# Improvement of Hydrogen Production during Anaerobic Fermentation of Food Waste Leachate by Enriched Bacterial Culture Using Biochar as an Additive

**DOI:** 10.3390/microorganisms9122438

**Published:** 2021-11-26

**Authors:** Van Hong Thi Pham, Jaisoo Kim, Soonwoong Chang, Woojin Chung

**Affiliations:** 1Department of Environmental Energy Engineering, Graduate School, Kyonggi University, Suwon 16227, Korea; vanhtpham@gmail.com; 2Department of Life Science, College of Natural Science, Kyonggi University, Suwon 16227, Korea; jkimtamu@kyonggi.ac.kr; 3Department of Environmental Energy Engineering, College of Creative Engineering, Kyonggi University, Suwon 16227, Korea

**Keywords:** hydrogen-producing bacteria, food waste leachate treatment, enriched bacteria, waste biodegradation

## Abstract

It has become urgent to develop cost-effective and clean technologies for the rapid and efficient treatment of food waste leachate, caused by the rapid accumulation of food waste volume. Moreover, to face the energy crisis, and to avoid dependence on non-renewable energy sources, the investigation of new sustainable and renewable energy sources from organic waste to energy conversion is an attractive option. Green energy biohydrogen production from food waste leachate, using a microbial pathway, is one of the most efficient technologies, due to its eco-friendly nature and high energy yield. Therefore, the present study aimed to evaluate the ability of an enriched bacterial mixture, isolated from forest soil, to enhance hydrogen production from food waste leachate using biochar. A lab-scale analysis was conducted at 35 °C and at different pH values (4, no adjustment, 6, 6.5, 7, and 7.5) over a period of 15 days. The sample with the enriched bacterial mixture supplemented with an optimum of 10 g/L of biochar showed the highest performance, with a maximum hydrogen yield of 1620 mL/day on day three. The total solid and volatile solid removal rates were 78.5% and 75% after 15 days, respectively. Acetic and butyrate acids were the dominant volatile fatty acids produced during the process, as favorable metabolic pathways for accelerating hydrogen production.

## 1. Introduction

The development of clean, sustainable energy has attracted significant attention in the recent decades, owing to severe environmental pollution, the gradual depletion of fossil fuels, and excessive greenhouse gas emissions. Hydrogen is considered to be the most promising alternative fuel, as it provides cost-effective and non-polluting energy. Compared to physicochemical processes, the biological production of hydrogen is more attractive, as it can be performed at ambient temperatures and pressures, with less energy consumed, and is more environmentally friendly [1,2]. Among the various biological strategies for H_2_ production, the anaerobic fermentation of rich carbon sources is the most practically applicable to achieve both waste degradation targets and high H_2_ yields.

Food waste leachate (FWL) is a major secondary wastewater pollutant generated by FW during the decomposition process; however, studies on bioenergy production from FWL have been limited. This wastewater is preferentially treated by anaerobic fermentation (AF), due to its high biodegradability and moisture content, to produce important products, such as volatile fatty acids (VFAs) and hydrogen [3,4]. Although advances in AF technologies have been made to better utilize the capacity of resource conversion from FWL, the field continues to face numerous limitations, such as reaction rates and substrate mass transfer during AF. The efficiency of mass transfer depends on the concentration gradient established in heterogeneous systems, due to the different sources of food waste [5,6]. Additionally, high concentrations of metabolic intermediates in the FWL, such as VFAs, are toxic to certain bacteria, and affect specific enzymes involved in the hydrolysis and acidogenesis processes [7].

Over the past decade, most studies have focused on enhancing the yield of H_2_ by optimizing operational parameters, such as the following: pH, temperature, total solid content (TS), and C:N ratio, to increase the digestibility of FW [8]. Other studies on the production of H_2_ from cellulose-based biomass have addressed the pretreatment steps [9,10]. Pretreatment methods can be classified as physical, chemical, and biological [11]; however, these methods can increase the overall time and cost of the process. Therefore, bioprocessing using bacteria is an alternative to the conversion of carbohydrates and other organic compounds into H_2_ [12]. Previous studies have investigated a wide range of H_2_-producing bacteria, including species in the genera *Rhodobacter, Rhodopseudomonas*, and *Rhodospirillum* [13,14]. Among these, *Rhodobacter capsulatus* is an ideal candidate, owing to its high conversion efficiency and ability to utilize various substrates for its growth [15]. In another study, H_2_ production was optimized by a co-culture of *Clostridium acidisoli* and *Rhodobacter sphaeroides* [16]. Furthermore, H_2_ is a by-product of complex bacterial metabolic pathways, attracting more in-depth studies on the microbial ecology of H_2_ production in mixed cultures.

In the recent years, carbon-based materials have been considered to improve the efficiency of resource recovery to biogas by enhancing microbial activity. Biochar has been investigated as one of the most promising materials, owing to its various sources and low cost of production [17,18,19,20,21]. Recent studies have reported an enhancement in biogas yield from the AD of food waste with the addition of biochar [22,23,24,25]. The food waste produced higher methane, from 55% to 78%, with the addition of 8.3 g/L biochar [26]. Other studies also showed the role of biochar in increasing methane production from mixed kitchen waste, by the addition of vermicompost-based biochar [27]. However, the performance of biochar for FWL treatment and hydrogen production from FW has yet to be reported. Therefore, this study aimed to examine the efficiency of H_2_ production from FWL using an active mixture bacterial culture in addition to biochar under various initial pH values, without previous pretreatment.

## 2. Materials and Methods

### 2.1. Feedstock and Enriched Bacterial Cultures

The FWL was collected from the Suwon Environment Affairs Agency, South Korea, and stored in a refrigerator at 4 °C until further use. Biochar (B) used as an additive in this study was treated with chaff charcoal purchased from Yougi Industry Company, and was subsequently ground and sieved to grains 1–2 mm in size before use. The physical and chemical properties of the original FWL and biochar were analyzed using standard methods [28].

The enriched bacterial culture was obtained during the development of a new culturing method for unculturable bacteria from various environments. In this study, a soil sample collected from the root-surrounding area of the forest soil at Kyonggi University, Suwon, South Korea, was used as the bacterial isolation source. The medium used in this study was the same as that described in previous studies with the following composition (per 1 L): glucose 10 g, beef extract 2 g, peptone 4 g, yeast extract 1 g, NaCl_2_ 5 g, K_2_HPO_4_ 1.5 g, MgCl_2_·6H_2_O 0.6 g, FeSO_4_·7H_2_O 0.2 g, L-cysteine 0.5 g, trace elements 10 mL, and vitamin solution 10 mL. The vitamin solution contained the following (per 1 L): riboflavin 0.025 g, citric acid 0.02 g, folic acid 0.01 g, and para-aminobenzoic acid 0.01 g. The trace element solution was composed of the following (per 1 L): MnSO_4_·7H_2_O 0.01 g, ZnSO_4_·7H_2_SO_4_ 0.05 g, H_3_BO_3_ 0.01 g, N(CH_2_COOH)_3_ 4.5 g, CaCl_2_·2H_2_O 0.01 g, Na_2_MoO_4_ 0.01 g, CoCl_2_·6H_2_O 0.2 g, and AlK(SO_4_)_2_ 0.01 g [4,29]. The vitamin solution was added after autoclaving the medium and trace elements as the last step. Five grams of soil was added to 50 mL of the above medium. After a 7-day incubation period, serial dilutions were spread onto agar medium plates including carboxymethylcellulose agar (CMCA, Sigma, Seoul, Korea), modified agar containing (per 1 L) 5 g of peptone, 5 g of yeast extract, 5 g of NaCl, and 15 g of agar, all of which were then incubated under both aerobic and anaerobic conditions for 7 days at 30 °C. All facultative colonies, which formed a clear zone on each agar plate after incubating at 30 °C for 48 h using iodine as an indicator for CMC degradation, were recorded as positive activities. The three bacterial strains (V1, V2, and V3) with the strongest enzyme activities were then examined in subsequent studies for anaerobic digestion and H_2_ production in the present study.

### 2.2. Experimental Set-Up and Operation

The biodegradation of FWL and H_2_ production was performed in 500 mL serum bottles capped with natural rubber sleeve stoppers. Each trial included 300 mL of FWL, 5% mixed bacterial cultures, 5–20 g/L of biochar (5 unit intervals), and an initial pH ranging from 4.5 to 7.5 at different temperatures between 30 and 55 °C (5 °C intervals). Each control was set up identically for all experiments to ensure proper comparison. Finally, all trials were purged with N_2_ gas for 5 min to maintain anaerobic conditions. The chemical parameters and gas produced in each bottle were measured regularly for 15 days to determine the optimal conditions for this process. All experiments were performed in triplicates.

### 2.3. Analytical Methods

#### 2.3.1. Physico-Chemical Analysis

Total solids (TS) and volatile solids (VS) were determined following the standard methodology of the American Public Health Association [28]. The pH of each sample was determined using a pH meter (HI 2210). Samples were centrifuged for 20 min and subsequently filtered through a 0.45 μm membrane filter for further analysis. The total chemical oxygen demand (TCOD) and soluble chemical oxygen demand (SCOD) were measured using an LCK 514 COD cuvette and a Hach Lange DR 5000 spectrophotometer (HACH EUROPE, Hach Lange, Germany); other samples were adjusted to pH 2.5 for VFA analysis. Following extraction with ether, the VFA concentration and composition were determined by gas chromatography (GC-6890N, Agilent Inc., Wilmington, DE, USA). The daily biogas volume was measured by BIOGAS 5000 (Geotechnical Instruments UK Ltd., Warwickshire, UK) during this study period. The C:N ratio was determined based on the values of total carbon (TC) and total nitrogen (TN), both of which were measured using a TruSpec CN carbon/nitrogen determinator (LECO Corporation, St. Joseph, MI, USA) [30].

#### 2.3.2. Biodiversity and Phylogenetic Analysis of Promising Bacterial Strains

Microbial analysis was performed by the plate counting method using nutrient agar (NA, BD Difco™, Becton, Dickinson and Company, Sparks, MD, USA) to investigate the total number of bacteria; a medium containing gelatin, standard salt solution, and trace element solution for proteolytic bacteria; actinomyces agar for actinomycetes; carboxymethylcellulose agar for cellulolytic bacteria; potato dextrose agar (PDA, BD Difco™). One milliliter of sample was collected from each trial after one, three, five, seven, ten, and fifteen days, serially diluted from 10^−1^, 10^−2^, 10^−3^, 10^−5^ to 10^−6^, and then 100 μL of each sample was spread and inoculated into different bacterial culture media. Petri dishes were incubated at 28 °C for 15 days. Colonies that appeared on the agar plates were enumerated as CFU/g of the liquid sample. Each pure colony was used to degrade organic compounds, including proteins from skim milk agar (Sigma-Aldrich, St. Louis, MO, USA), complex polysaccharides from starch (Sigma-Aldrich), and cellulose (Sigma-Aldrich) at a range of pH values (4.5–7.5) and temperatures (30–55 °C). Strongly active bacterial strains were prepared for further analysis.

To determine partial 16S rRNA sequences, genomic DNA from all bacterial strains was extracted according to a previously published method [31]. The 16S rRNA gene was amplified using the 27F and 1492R universal primers, and sequencing was performed by Macrogen using the 785F and 907R primers [32]. A multiscreen filter plate (Millipore Corp, Bedford, MA, USA) was used to purify the PCR products, which were then sequenced using 518F (50-CCAGCAGCCGCGGTAATACG-30) and 800R (50-TACCAGGGTATCTAATCC-30) primers with the PRISM BigDye Terminator v3.1 Cycle Sequencing Kit (Applied Biosystems, Foster City, CA, USA). This process was conducted at 95 °C for 5 min, after which the product was cooled on ice for 5 min, and then analyzed using an ABI Prism 3730XL DNA analyzer (Applied Biosystems, Foster City, CA, USA). SeqMan software (DNASTAR Inc., Madison, WI, USA) was used to assemble the nearly full-length 16S rRNA sequences. This sequence was compared with those of other related bacterial strains using the EZBioCloud server (http://ezbiocloud.net, accessed on 8 October 2021) for identification [33].

## 3. Results

### 3.1. Phylogenetic Analysis

The three bacterial strains (V1, V2, and V3) identified through 16S rRNA sequencing were found to belong to the genera Rossellomorea and Bacillus. They were the closest to Rossellomorea oryzaecorticis R1^T^, Bacillus velezensis CR-502^T^, and Bacillus albus, with similarities of 67%, 99.93%, and 100%, respectively.

### 3.2. Physiochemical Characteristics and Hydrolysis Performance

The original FWL and biochar were first characterized before the experiment, with each property shown in Table 1. Figure 1a and Table 2 show the removal efficiencies of TS, VS, and COD. The TS removal rate was 55%, 65%, 70%, and 78.5%, whereas that of VS was 60%, 63%, 68%, and 75%, for the control, control with biochar, mixed bacterial culture (MBC) only, and MBC with 10% biochar (pH 5.5 after adding biochar), respectively. Enrichment of the bacterial mixture, with the addition of 10% biochar, illustrated the efficiency of the hydrolysis yields, with a significant increase of 640 gSCOD/kg of VS added. These experiments were carried out under the optimal conditions of pH 6.5 and 35 °C.

To evaluate the potential of raw FWL for H_2_ and VFA production more efficiently, the total VFA yield was determined. Figure 1a,b show the optimal point of the total VFA on day three of incubation at 35 °C as the optimal temperature, with 9850 mg/L. Simultaneously, acetic acid was observed to be the dominant VFA, reaching up to 4850 mg/L, followed by butyric acid and propionic acid, accounting for 3350 and 1650 mg/L, respectively. The concentration of butyric acid peaked at 3670 mg/L on day eight. It was clear that acetic acid and butyric acid were produced stably from days three to nine. The total VFA slightly decreased until day 10, and subsequently underwent a rapid decrease to 1050 mg/L on day 15.

### 3.3. Effect of pH on Hydrogen Production

Figure 2 illustrates the H_2_ yield efficiency at different pH values for the four trials. At pH 4, low hydrogen production was observed in all four trials, as shown by the following results: 1140 mL from the MBC with biochar, 950 mL from the MBC without biochar, and 890 mL and 750 mL for the control with and without biochar, respectively. The trend increased from the pH with no adjustment to pH 6 for all the samples. The amount of H_2_ in the MBC with biochar was the highest at 1290 mL, followed by 1220 mL for the control with biochar. When the pH was 6.5, the optimal values for H_2_ production of the control and MBC with biochar were 1150 mL and 1620 mL, respectively. A decreasing trend was observed when the pH increased, except for the H_2_ from the MBC without biochar, which remained stable up to pH 7. In general, the MBC with biochar exhibited the highest H_2_ production performance compared to other samples during the process.

Figure 3 shows the H_2_ yield at 35 °C for 10 days, at different pH values of the MBC with biochar, in comparison with the timing of each VFA produced per day. The amount of H_2_ reached its highest concentration on day three at all pH values, accounting for 1620 mL at pH 6.5, and was stable on day four, except for the pattern at pH 7, which started to decrease after day three. After 7 days, the H_2_ at a pH of 6.5 only increased again on day eight, after a moderate reduction, and reached a concentration of 1450 mL before gradually reducing. An acidic pH of 4 or pH 7 are unfavorable conditions for H_2_-producing bacteria.

The four trials were incubated at a no-adjustment pH for 10 days, and H_2_ production was measured regularly (see Figure 4). Samples in the presence of the MBC both with and without biochar exhibited enhanced H_2_ production, with peaks at 1380 mL and 1050 mL on day three, respectively. The H_2_ yield decreased slightly between days three and five, before decreasing sharply thereafter. The control with biochar reached its highest concentration of 880 mL on day six, and 820 mL on day seven for the control without biochar.

The hydrogen production rate and hydrogen yield of each MBC with different biochar ratios, ranging from 5 to 20 g/L (5 unit interval), are shown in Figure 5. The highest amount of hydrogen was generated on day three, reaching up to 1350 mL/day at a 10 g/L biochar addition rate. However, the sample with 20 g/L of biochar added peaked in hydrogen production earlier on day two, and decreased after that. The data showed that biochar ratios lower or higher than 10 g/L yielded less hydrogen.

### 3.4. Microbial Analysis

Figure 6 shows the difference in microbial groups in the four trials fermented at 35 °C, pH 6.5 on day three. Numerous bacterial populations contributed to the hydrolysis and acidogenesis of FWL. Based on taxonomic classification, 15 different genera that belong to three phyla (*Firmicutes*, *Proteobacteria,* and *Actinobacteria*) were identified. Eight genera were found in the control sample, seven in the control with biochar, and 12 in the MBC without biochar, but all 15 genera were found in the MBC with biochar. *Bacillus* was the dominant genus in all four trials and accounted for the highest proportion (45%) of enriched bacterial mixture without biochar. Additionally, this genus was identified in 44% of the MBC with biochar, and 40% and 0.35% for the control with biochar and control only, respectively. *Paenibacillus* and *Brevibacillus* were the second most well-represented genera in all four trials. The genus *Enterobacter* was dominant in the control and the control with biochar at 17% and 18%, respectively, and 2% and 4% in the MBC with and without biochar, respectively. *Clostridium* was found in 4% and 6% of the controls with and without biochar, but this number was lower than the 2% and 3.5% in the samples inoculated with MBC.

## 4. Discussion

The VS/TS ratio was 91% of the high organic content, illustrating its high potential for biodegradation [34]. The C:N ratio of the FWL in this study was 22.5, which was in line with the findings of Kondusamy and Kalamdhad (2014) [35], who reported that a C:N ratio of 20–30 was sufficient for the AD process.

This was similar to a previous study, which reported that a higher TS content supplied more nutrients for microbial growth, which may enhance the performance of the whole process [36]. However, H_2_ production may decrease once it exceeds the optimal point. Other studies have reported a decrease in H_2_ production with an increase in TS content [37,38]. The 16% TS used in this study was defined as near to the point of dry fermentation, and as a great condition for acetic acid and butyric acid production [39,40].

Hydrogen production even occurred under acidic conditions (pH 4) in all four samples, similarly to a previous study, which concluded that the performance of the first phase was better at an acidic pH [41]. Some studies suggested that neutral and alkaline conditions provide the optimal pH; however, in this study, pH 6.5 was observed to be the optimal pH for H_2_ generation [42,43,44]. Furthermore, at an initial pH higher than 7, the conditions for microbial activity lead to rapid acid accumulation. Therefore, a pH of 6–6.5 is favorable for the growth and longer, more efficient activity of H_2_-producing bacteria; as such, Sunyoto et al. found the maximum H_2_ production at pH 6 with 820 mL/day [41]. However, there is no definite point at which the initial pH H_2_ production would reach the maximal value, due to complex factors, such as operational parameters, inoculum in the systems, and other factors during the fermentation process.

H_2_ production depends on each VFA produced according to the following pathways by acetic acid bacteria:C_6_H_12_O_6_ + 2H_2_O ⟶ 2CH_3_COOH + 4H_2_ + 2CO_2_ (acetate fermentation)

When acetic acid production started to decrease, the butyric acid yield increased rapidly after day three. Hydrogen production peaked again on day eight, when butyric acid was the most abundant VFA in the contribution of butyrate-producing species. The reaction can be expressed as follows:C_6_H_12_O_6_ ⟶ 2CH_3_CH_2_CH_2_COOH + 2H_2_ + 2CO_2_ (butyrate fermentation)

The butyrate fermentation pathway contributes more to H_2_ production than acetate, as the latter can be produced by homoacetogenesis [45]. When the concentration of acetic acid increased and less H_2_ was produced, microorganisms utilized alternate metabolic pathways, such as the following, by H_2_-consuming microbes:2CO_2_ + 4H_2_ ⟶ CH_3_COOH + 2H_2_O (homoacetogenesis)
C_6_H_12_O_6_ ⟶ 3CH_3_COOH (homoacetogenesis)

The H_2_ that is produced may be consumed by propionic-producing bacteria to release propionic acid according to the following pathways:C_6_H_12_O_6_ + 2H_2_ ⟶ 2CH_3_CH_2_COOH + 2H_2_O (propionate fermentation)

Therefore, the accumulation of VFA may inhibit H_2_ generation.

However, the optimal conditions for H_2_ and VFA generation may be different. The decrease in pH, caused by VFA accumulation, resulted in lower H_2_ production [46].

It is believed that the biochar with an initial pH higher than 8.0 present in the samples acts as a buffer to prevent an enormous pH decrease, which may promote a higher H_2_ yield due to system maintenance [41,47]. Furthermore, it is known that biochar with 10% volatile matter may serve as a nutrient supplement for microorganisms [48]. However, there are limited studies on the effects of biochar on FWL treatment. This study is the second of its kind, after a previous study that was carried out with an addition of 10% biochar, but without inoculums, which obtained a maximum H_2_ volume of 820 mL/day, which was less than the H_2_ amount generated in this study with 1620 mL/day [41]. Here, we demonstrated that H_2_ production that was significantly enhanced by enriched bacterial culture reached a peak of 1620 mL of H_2_ per day.

There was a difference in the dynamic microbial community between the control group with and without enriched bacterial inoculum. As shown in Figure 6, more genera were present in the inoculated samples, both with and without biochar addition. However, the microbial community driver in the best samples included bacteria and biochar with 15 genera. This may indicate the strong effect of biochar on microbial activities, which may be investigated in future studies. *Clostridium* was the only dominant genus initially, while the genus *Lactobacillus* dominated during digestion, and the *Bacillus* in this study was dominant throughout the process in all four sets of samples [49,50,51]. In particular, the genus *Rossellomorea* only presented with the addition of MBC, with 1% in MBC only and 3% in the MBC-added biochar. However, to understand the change in the microbial community after adding MBC, the follow-up experiments should be examined to investigate how each species in the MBC individually contributes to hydrogen generation from FWL.

The new method proposed in this study improved the efficiency of food waste leachate fermentation using enriched bacterial inoculum and biochar for 15 days. This approach shortened the fermentation time 2–5-fold compared to other methods, due to the strong activities of the microbes [52]. However, this study does not have enough data to figure out the metabolic pathways in each trial, which change the efficiency of the whole process. Therefore, other experiments should be performed, in addition to each bacterial strain individually, at different temperatures and various biochar concentrations. Moreover, the question of how biochar supports biofilm formation and provides the temporary nutrients that are beneficial to enhance hydrogen production needs to be answered in a further study.

## 5. Conclusions

The use of biochar in this study showed a contribution to the enhancement of H_2_ production in the presence of enriched bacterial mixed culture at an optimal pH of 6.5, to obtain 1620 mL of H_2_ per day. Hydrogen production peaked on day three of the experimental process, and remained stable for 3–8 days before decreasing. Herein, we introduce a useful bacterial mixture, in addition to biochar, to improve H_2_ production and the removal of organic matter from food waste leachate, which may benefit wastewater treatment plants and inspire the investigation of novel functional bacteria from the environment. Such hydrogen production, with suitable biological process technologies in the waste-to-energy approach, is still a valuable and viable alternative method for researchers to investigate in the future. However, further studies are necessary to understand the mechanisms and microbial activities of each group in the presence of biochar for each bacterial strain, V1, V2, and V3 individually. The role of trace elements and biochar in the AD of food waste, independently, mostly in single-stage anaerobic systems, was evaluated. However, there is no study to address the role of biochar in improving the bioavailability of trace elements in AD. Therefore, the effect of biochar in the waste treatment process is still questionable, and should be explored in further studies.

## Figures and Tables

**Figure 1 microorganisms-09-02438-f001:**
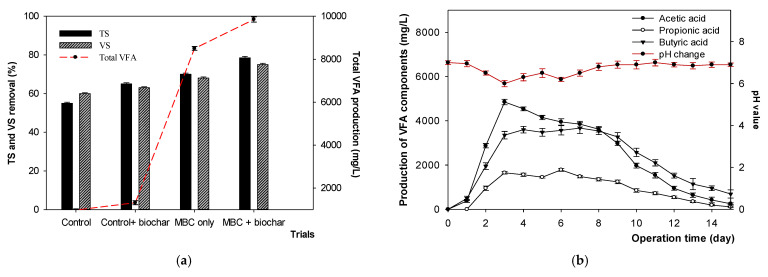
(**a**) The efficiency of TS, VS removal and production of total VFA, and (**b**) concentration of the main VFAs produced at different pH values during 15 days of incubation at 35 °C.

**Figure 2 microorganisms-09-02438-f002:**
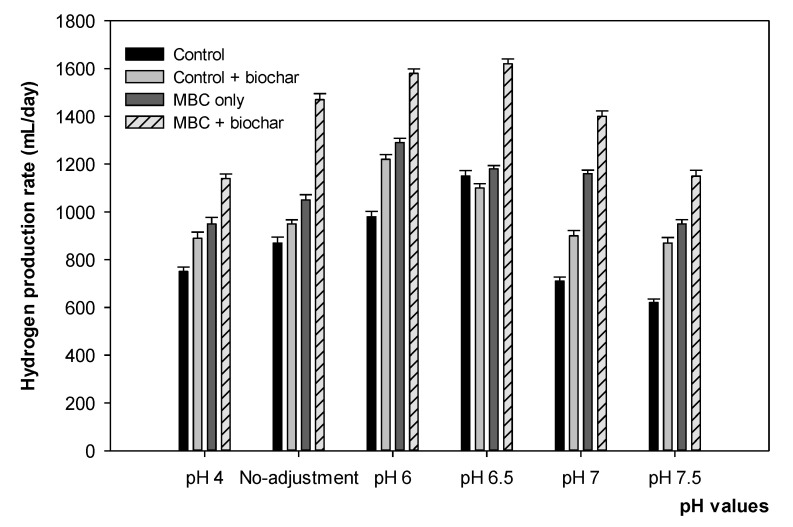
The effect of pH values on hydrogen production of four trials in this study at 35 °C. After adding 10% biochar, pH 5.5 was the final value of no-adjustment trial.

**Figure 3 microorganisms-09-02438-f003:**
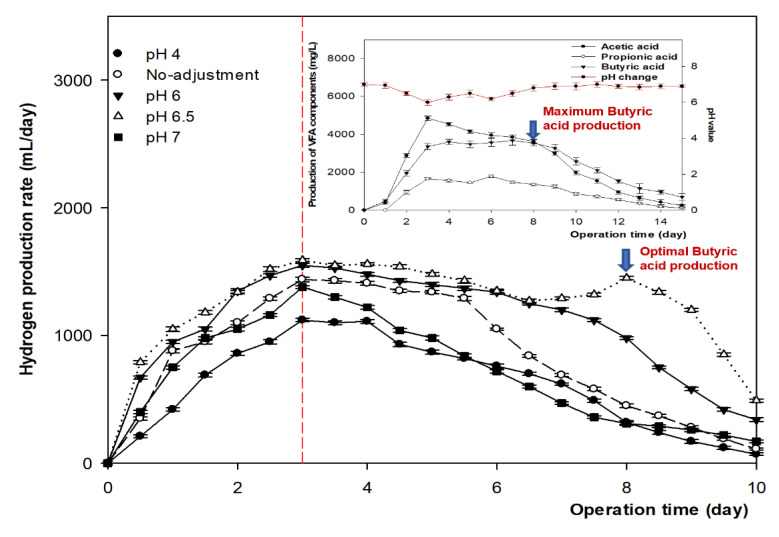
Hydrogen production of the MBC with biochar at different pH values in 10 days at 35 °C.

**Figure 4 microorganisms-09-02438-f004:**
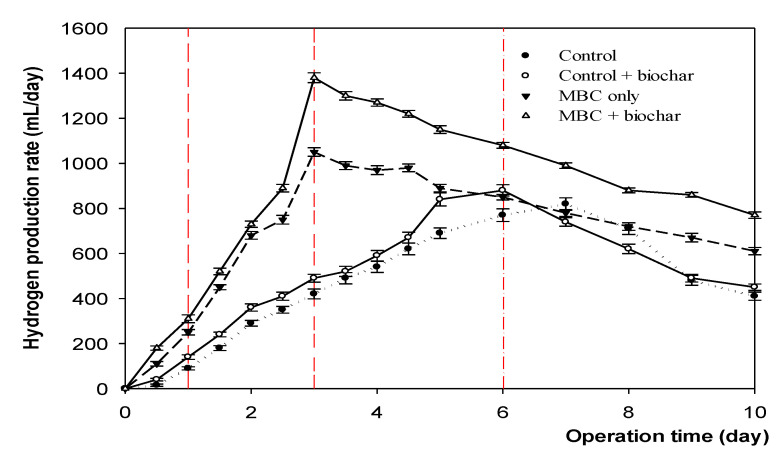
The hydrogen production of four trials was performed at 35 °C, with no pH adjustment for 10 days.

**Figure 5 microorganisms-09-02438-f005:**
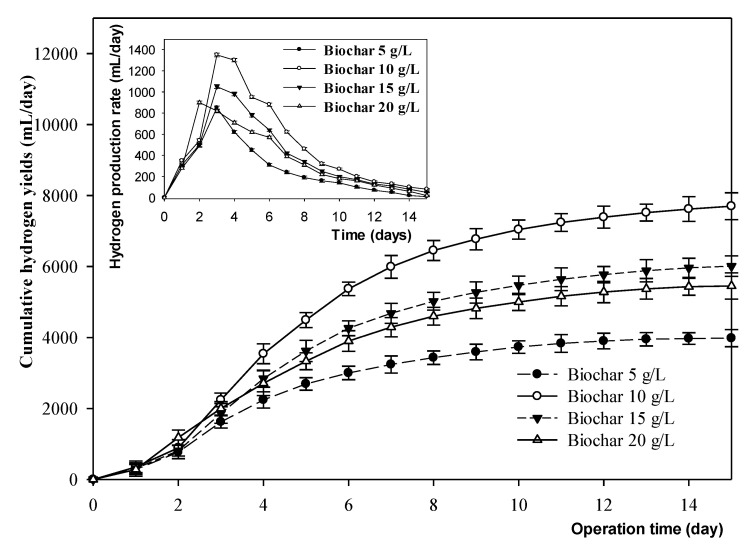
Production rate and cumulative yield of hydrogen with different biochar ratios of the MBC. This experiment was tested at 35 °C with no pH adjustment for 15 days.

**Figure 6 microorganisms-09-02438-f006:**
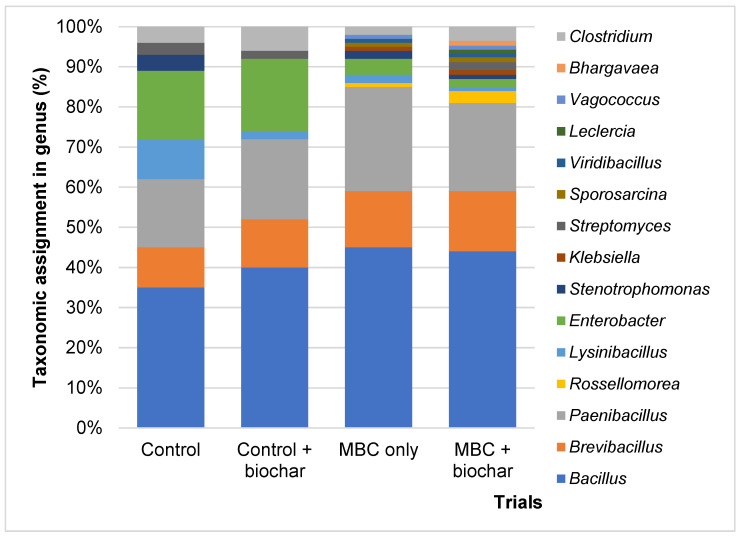
The microbial composition of four trials at pH 6.5, 35 °C on day 3.

**Table 1 microorganisms-09-02438-t001:** Characteristics of the original food waste leachate and biochar used in this study.

Parameters	Food Waste Leachate	Biochar
Total solids (TS%)	16 ± 0.35	ND *
Volatile solids (VS%)	14.6 ± 0.22	ND *
VS/TS (%)	91 ± 0.45	3.68 ± 0.1
Moisture content (MC%)	84 ± 0.55	30.8 ± 0.52
TBOD_5_ (g/L)	146.1 ± 1.4	ND *
SBOD_5_ (g/L)	57.87 ± 0.62	ND *
TCOD (g/L)	161.17 ± 0.94	ND *
SCOD (g/L)	80.41 ± 0.39	ND *
Total Nitrogen (TN)	5.28 ± 0.15	0.14 ± 0.013
Total Phosphorus (TP)	0.88 ± 0.02	44.7 ± 0.37
Total carbon	45 ± 0.26	44.7 ± 0.44
pH	5 ± 0.19	8.2 ± 0.14
C:N ratio	22.5 ± 0.24	317.9 ± 2.2

ND *: not determined.

**Table 2 microorganisms-09-02438-t002:** Performance of fermentation process obtained from four samples at optimal conditions.

Parameters	Control	Control with Biochar	MBC Only	MBC with Biochar
Acetate (mg/L)	540 ± 5.3	820 ± 11.2	3800 ± 59.5	4850 ± 38.8
Propionate (mg/L)	0	20 ± 2.4	1250 ± 9.4	1780 ± 13.5
Butyrate (mg/L)	440 ± 10.6	480 ± 22.2	3450 ± 12.9	3670 ± 40.2
Total VFA production (mg/L)	980 ± 56.7	1320 ± 90.5	8500 ± 88.7	9850 ± 120.4
Hydrogen yields (mL/day)	1150 ± 20.7	1220 ± 30.1	1290 ± 41.3	1620 ± 30.5

## Data Availability

The data used to support the findings of this study are available from the corresponding author upon request.

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
