# Peer review of "Improvement of Hydrogen Production during Anaerobic Fermentation of Food Waste Leachate by Enriched Bacterial Culture Using Biochar as an Additive"

_microorganisms, 2021, doi:10.3390/microorganisms9122438_

Round 1
Reviewer 1 Report
The paper describes a systematic set of experiments performed to measure the hydrogen and volatile fatty acid production from food waste leachate at a range of initial pH values either with or without the addition of biochar or a mixture of three bacterial strains isolated from a previous experiment. The authors found the highest hydrogen production yield from fermentations with an initial pH 6.5 supplemented with 10g/L of biochar and the mixture of bacterial strains. The combined set of results provide an example that the addition of biochar to food waste leachate can enable additional hydrogen production by added bacteria.
The presentation of the Results section is difficult to follow and hinders the message of the paper. The identification of the three bacterial species used in the “mixture” without context in the Results for how they were identified or why they were used is disorienting. Re-working the Results section to address this point and providing additional clarity on the number of replicates used and what the error bars show will strengthen the quality of the paper. There are also some potential English translation issues in the Results section that make it unclear if experiments were performed once for four different conditions/samples, or using four replicates for each of the four conditions/samples. Ideally, the second is true and only clarification of the language is needed, as it is important to perform these experiments for more than one replicate of each condition/sample. Additional specific comments are below:
- Axis titles and labels on figures are difficult to read; the font size should be increased.
- For all figures, information should be provided with respect to what type of error is shown by the error bars, e.g. standard deviation or standard error of the mean.
- The insets in Figures 3 and 5 are difficult to read and should be presented as separate sub-figures.
- Line 85, change from “medium was used in this study” to “medium used in this study”.
- Line 105, in the Materials and Methods it is stated that “initial pH ranging from 5 to 10” was used for hydrogen production experiments. In Figure 2, data for pH 4.5 to pH 7.5 is presented. No data is presented for pH values above 7.5. The values in the Materials and Methods section should be adjusted from “5 to 10” to “4.5 to 7.5”.
- Line 131, replace “tomato” with “potato”.
- Section 3.1, additional context should be provided in this section with respect to how bacterial strains V1, V2, and V3 were isolated and identified for use in further experiments.
- Table 1, no information is provided for the meaning of “ND*”. Presumably no data was collected, which should be noted in the table legend.
- Section 3.2, the “mixture”/”bacterial mixture” should be defined. Presumably these are the 3 strains described in Section 3.1 and could be defined as such there.
- Section 3.2, the amount of biochar used for the experiment described in Figure 1 is not defined.
- Line 178, “until the 10-day quarantine…” The intent of this phrase is unclear, but the use of the word quarantine is incorrect.
- Table 2, units should be provided for acetate, propionate, and butyrate amounts.
- The TS%, VS%, and Total VFA production presented in Table 2 are redundant with Figure 1(a) and should only be presented in one of these locations.
- The error bars presented in Figure 1(a) are substantially larger than the numerical error presented in Table 2, which of these is correct?
- Figure 1, error bars are shown in Figure 1(a) for some measurements, but not for the VFA production or for any measurements in Figure 1(b). Authors should include error bars for all plotted data and indicate the number of replicates used in the experiment.
- Section 3.2/Figure 2, the usage of the words “samples” and “trials” are ambiguous here and elsewhere. These should be changed to “replicates” or “biological replicates”, if biological replicates were used.
- Figure 2, the pH of the biochar was measured as 8.2 in Table 1. It should be made clearer what the initial pH of the “No-adjustment with biochar” media was, as it is unlikely to have remained 5. Similarly, it should be made clear that the pH of the media was adjusted to the other stated pH values after the addition of the biochar. The amount of biochar used in these experiments should also be noted.
- Figure 2, the connection of the presented data points with lines is misleading, as they don’t connect the same samples that had their pH changing, but rather independent experiments carried out at distinct starting pH. The lines should be removed and the data should be presented either as the points alone or as a bar graph.
- Figure 6/Discussion, The authors should comment on the presence/amount of the Rossellomorea present in the “mixture” and “mixture with biochar” samples, as this species is clearly only present due to its addition in the mixture. As it is present in a very small amount in both cases and the added Bacillus species cannot be distinguished from the background Bacillus species in the FWL, a good follow-up experiment would be to examine how each of these species contributes to the hydrogen/VFA production from the FWL individually.
- Line 302 & line 316, the word “significant” should not be used unless a statistical test is conducted to demonstrate that a significant difference exists between two sets of data.
Author Response
Dear Reviewer,
We would like to thank you very much for your valuable time with the helpful comments on my manuscript. We have responded to each comment point by point in the attachment. We hope that you will satisfy with this revision.
All the best,
Woojin Chung
Dear Chief Editor and Reviewer,
First of all, we would like to thank you for your insightful comments and valuable suggestions on our manuscript. We have revised the manuscript following each comment. We hope that our response of questions listed as below will make you satisfied.
Comments and Suggestions for Authors
The paper describes a systematic set of experiments performed to measure the hydrogen and volatile fatty acid production from food waste leachate at a range of initial pH values either with or without the addition of biochar or a mixture of three bacterial strains isolated from a previous experiment. The authors found the highest hydrogen production yield from fermentations with an initial pH 6.5 supplemented with 10g/L of biochar and the mixture of bacterial strains. The combined set of results provide an example that the addition of biochar to food waste leachate can enable additional hydrogen production by added bacteria.
The presentation of the Results section is difficult to follow and hinders the message of the paper. The identification of the three bacterial species used in the “mixture” without context in the Results for how they were identified or why they were used is disorienting. Re-working the Results section to address this point and providing additional clarity on the number of replicates used and what the error bars show will strengthen the quality of the paper. There are also some potential English translation issues in the Results section that make it unclear if experiments were performed once for four different conditions/samples, or using four replicates for each of the four conditions/samples. Ideally, the second is true and only clarification of the language is needed, as it is important to perform these experiments for more than one replicate of each condition/sample. Additional specific comments are below:
Point 1: Axis titles and labels on figures are difficult to read; the font size should be increased.
Response 1: Thank you very much for your useful recommend on this part. I have revised all pictures to make them clear.
Point 2: For all figures, information should be provided with respect to what type of error is shown by the error bars, e.g. standard deviation or standard error of the mean.
Response 2: We are appreciated at your helpful comment. We have reworked to show the error bars on each graph.
Point 3: The insets in Figures 3 and 5 are difficult to read and should be presented as separate sub-figures.
Response 3: Thank you very much for your useful point of view. We revised and use the large font size that makes those Figures better than before. We hope that will make you satisfied at this point.
Point 4: Line 85, change from “medium was used in this study” to “medium used in this study”.
Response 4: Thank you very much for your comment for this grammar mistake. We have corrected it (line 97).
Point 5: Line 105, in the Materials and Methods it is stated that “initial pH ranging from 5 to 10” was used for hydrogen production experiments. In Figure 2, data for pH 4.5 to pH 7.5 is presented. No data is presented for pH values above 7.5. The values in the Materials and Methods section should be adjusted from “5 to 10” to “4.5 to 7.5”.
Response 5: Thank you very much for your helpful point on this mistake of the manuscript. We have revised it already (line 120)
Point 6: Line 131, replace “tomato” with “potato”.
Response 6: Thank you very much for your useful comment. We have corrected it (line 147).
Point 7: Section 3.1, additional context should be provided in this section with respect to how bacterial strains V1, V2, and V3 were isolated and identified for use in further experiments.
Response 7: Thank you very much for your helpful comment. We have rewritten a clear paragraph in section 2.1 (line 105-115). We hope that this revision will make you satisfied.
Point 8: Table 1, no information is provided for the meaning of “ND*”. Presumably no data was collected, which should be noted in the table legend.
Response 8: Thank you very much for your useful comment. We added the information this table (Line 189).
Point 9: Section 3.2, the “mixture”/”bacterial mixture” should be defined. Presumably these are the 3 strains described in Section 3.1 and could be defined as such there.
Response 9: Thank you very much for your helpful comment in this point. We have revised it as the Mixture bacterial culture (MBC) in the whole manuscript.
Point 10: Section 3.2, the amount of biochar used for the experiment described in Figure 1 is not defined.
Response 10: Thank you very much for your helpful review on this point. We have added more information in detail (Line 184)
Point 11: Line 178, “until the 10-day quarantine…” The intent of this phrase is unclear, but the use of the word quarantine is incorrect.
Response 11: Thank you very much for your kind comment. We revised this sentence already (Line 197).
Point 12: Table 2, units should be provided for acetate, propionate, and butyrate amounts.
Response 12: Thank you very much for your useful comment. We have added more information in the Table 2.
Point 13: The TS%, VS%, and Total VFA production presented in Table 2 are redundant with Figure 1(a) and should only be presented in one of these locations.
Response 13: Thank you very much for your kind comment on this point. We have deleted TS and VS % in Table 2.
Point 14: The error bars presented in Figure 1(a) are substantially larger than the numerical error presented in Table 2, which of these is correct?
Response 14: Thank you very much for your comment. We have corrected the error bars in Figure 1(a).
Point 15: Figure 1, error bars are shown in Figure 1(a) for some measurements, but not for the VFA production or for any measurements in Figure 1(b). Authors should include error bars for all plotted data and indicate the number of replicates used in the experiment.
Response 15: Thank you very much for your comment. We have fixed Figure 1(a) and 1(b).
Point 16: Section 3.2/Figure 2, the usage of the words “samples” and “trials” are ambiguous here and elsewhere. These should be changed to “replicates” or “biological replicates”, if biological replicates were used.
Response 16: We would like to thank you very much for your kind comment. We set up 4 trials with different components (1) Control-only FWL. (2) Control + biochar-FWL added biochar. (3) Mixed bacterial culture only-FWL added mixed bacterial culture. (4) Mixed bacterial culture + biochar-FWL inoculated mixed bacterial culture and biochar. Each of trials was set up in triplicate. Therefore, the use of “replicates” here may make the confusion to our triplicates.
I hope that you will satisfy with our explanation.
Point 17: Figure 2, the pH of the biochar was measured as 8.2 in Table 1. It should be made clearer what the initial pH of the “No-adjustment with biochar” media was, as it is unlikely to have remained 5. Similarly, it should be made clear that the pH of the media was adjusted to the other stated pH values after the addition of the biochar. The amount of biochar used in these experiments should also be noted.
Response 17: We would like thank you very much your useful comment in the significant point.
We have revised and added more information in the legend of Fig.2 (line 183,184; 217, 218…)
Point 18: Figure 2, the connection of the presented data points with lines is misleading, as they don’t connect the same samples that had their pH changing, but rather independent experiments carried out at distinct starting pH. The lines should be removed and the data should be presented either as the points alone or as a bar graph.
Response 18: Thank you very much for your kind comment. We have revised Figure 2 to the bar graph.
Point 19: Figure 6/Discussion, The authors should comment on the presence/amount of the Rossellomorea present in the “mixture” and “mixture with biochar” samples, as this species is clearly only present due to its addition in the mixture. As it is present in a very small amount in both cases and the added Bacillus species cannot be distinguished from the background Bacillus species in the FWL, a good follow-up experiment would be to examine how each of these species contributes to the hydrogen/VFA production from the FWL individually.
Response 19:
We would like thank you very much for your useful comment in this point. Following your opinion, we have revised the paragraph in Discussion (Line 327-331).
Point 20: Line 302 & line 316, the word “significant” should not be used unless a statistical test is conducted to demonstrate that a significant difference exists between two sets of data.
Response 20: Thank you very much for your kind comment. We have revised it (Line 319).
Reviewer 2 Report
The paper focused on hydrogen production using bacterial culture, investigating enriching methods using food wastes as substrate.
Few revisions are required and they are reported below:
- I suggest to revise the title
- figure 1 should be revised
- a nomenclature section should be added to the manuscript, including parameters and variables with proper SI unit of measure
- check that all details are added to section 2 for the instruments used
- add for table 1 a standard deviation value, or at least a plausible range of the measurements
- complete table 2 with the standard deviation values for all measurements
- figure 3,4 should be revised
- improve references for biochar uses [17] such as (10.1016/j.ijms.2018.05.002, 10.1016/j.renene.2021.01.147, 10.1016/j.renene.2019.03.077, 10.1016/j.scitotenv.2021.150531)
Author Response
Dear Reviewer,
We would like to thank you very much for your valuable time with the helpful comments on my manuscript. We have responded to each comment point by point in the attachment. We hope that you will satisfy with this revision.
All the best,
Woojin Chung
Dear Chief Editor and Reviewer,
First of all, we would like to thank you for your insightful comments and valuable suggestions on our manuscript. We have revised the manuscript following each comment. We hope that our response of questions listed as below will make you satisfied.
Comments and Suggestions for Authors
The paper focused on hydrogen production using bacterial culture, investigating enriching methods using food wastes as substrate.
Few revisions are required and they are reported below:
Point 1: I suggest to revise the title
Response 1: Thank you very much for your kind comment. We have revised the title of the manuscript. We hope that will satisfy you.
Point 2: figure 1 should be revised
Response 2: Thank you very much for your useful comment. We have revised it as added error bars, larger font size to make it clear and bigger.
Point 3: a nomenclature section should be added to the manuscript, including parameters and variables with proper SI unit of measure.
Response 3: Thank you very much for your useful comment. We have added the nomenclature section in the manuscript.
Point 4: check that all details are added to section 2 for the instruments used
Response 4: Thank you very much for your kind comment. We have checked all of them
(line 138, 140).
Point 5. add for table 1 a standard deviation value, or at least a plausible range of the measurements
Response 5: Thank you very much for your useful comment. We have added the standard deviation in Tabel 1.
Point 6: complete table 2 with the standard deviation values for all measurements
Response 6: Thank you very much for your useful comment. We have added the standard deviation in Tabel 2.
Point 7: figure 3,4 should be revised
Response 7. Thank you very much for your useful comment. I have revised Figure 3 and 4.
Point 8: improve references for biochar uses [17] such as (10.1016/j.ijms.2018.05.002, 10.1016/j.renene.2021.01.147, 10.1016/j.renene.2019.03.077, 10.1016/j.scitotenv.2021.150531)
Response 8: Thank you very much for your kind comment. We have added these references in the text as well as in the reference list as you recommended (ref.17-20). We hope that this will make you satisfied.
Reviewer 3 Report
Comments:
- Expand your abstract, please: what was the reason for the research and what is its purpose for science and industry. The same is true of the conclusions in this article.
2. The introduction lacks specific references and comparisons with the literature data, please complete this.
3. '….using a strong active mixture bacterial…’ – line 73, some of the terms as cited in this article are far too general and not substantive.
4. Extend the description in chapter 2.2. The content should be enriched with a diagram of the apparatus (if the title includes the wording lab-scale, it obliges you) and / or images, and / or a diagram of the process.
5. You did not specify the type and properties of the soil which the bacteria were isolated. I did not notice in the text the description about process isolation of bacterial cells from the soil.
6. Lines 293-295: explain why biochar is a buffer?
Author Response
Dear Reviewer,
We would like to thank you very much for your valuable time with the helpful comments on my manuscript. We have responded to each comment point by point in the attachment. We hope that you will satisfy with this revision.
All the best,
Woojin Chung
Dear Chief Editor and Reviewer,
First of all, we would like to thank you for your insightful comments and valuable suggestions on our manuscript. We have revised the manuscript following each comment. We hope that our response of questions listed as below will make you satisfied.
Comments and Suggestions for Authors
Comments:
Point 1: Expand your abstract, please: what was the reason for the research and what is its purpose for science and industry. The same is true of the conclusions in this article.
Response 1: Thank you very much for your useful recommend on this part. We have expanded the abstract and conclusion in yellow highlight (Line 14-20; 344-353).
Point 2: The introduction lacks specific references and comparisons with the literature data, please complete this.
Response 2: Thank you very much for your useful point of view. We have added more specific references (17-20, 25, 26).
Point 3 '….using a strong active mixture bacterial…’ – line 73, some of the terms as cited in this article are far too general and not substantive.
Response 3: We are appreciated at your useful comment. We have revised in the text (Line 81-84).
Point 4: Extend the description in chapter 2.2. The content should be enriched with a diagram of the apparatus (if the title includes the wording lab-scale, it obliges you) and / or images, and / or a diagram of the process.
Response 4: Thank you very much for your helpful comment. We have revised the title of manuscript.
Point 5: You did not specify the type and properties of the soil which the bacteria were isolated. I did not notice in the text the description about process isolation of bacterial cells from the soil.
Response 5: Thank you very much for your recommend as a significant point. We have added more information about soil and isolation process in the text (Line 94-96, 105-115).
Point 6: Lines 293-295: explain why biochar is a buffer.
Response 6: Thank you very much for your recommend. We added more information about pH value of biochar in the text (Line 310)
Round 2
Reviewer 1 Report
The authors adequately addressed concerns from the initial submission. The manuscript could still use some English language editing, but it is otherwise is acceptable for publication in its revised form.
Author Response
Dear Reviewer,
We would like to thank you very much for your valuable time and your insightful comments and valuable suggestions to improve our manuscript.
Best regards,
Woojin Chung
Reviewer 3 Report
Dear Authors,
I approve the revised version of the article and consent to the publication of the paper in its current form.
Regards,
Reviewer
Author Response

(The authors gave the same response as above.)
